# A Narrative Review Evaluating Diet and Exercise as Complementary Medicine for the Management of Alzheimer’s Disease

**DOI:** 10.3390/nu17111804

**Published:** 2025-05-26

**Authors:** Amanda N. Szabo-Reed, Mickeal N. Key

**Affiliations:** 1KU Alzheimer’s Disease Research Center, Fairway, KS 66205, USA; 2Department of Internal Medicine, Division of Physical Activity and Weight Management, University of Kansas Medical Center, Kansas City, KS 66160, USA; 3Department of Neurology, University of Kansas Medical Center, Kansas City, KS 66160, USA

**Keywords:** Alzheimer’s disease, diet, nutrition, exercise, physical activity, brain health, brain function, cognition

## Abstract

Alzheimer’s Disease (AD) is characterized by complex brain alterations leading to progressive cognitive decline and neuropsychiatric disturbances. This narrative review explores these changes and the potential of diet and exercise as modifiable lifestyle factors to mitigate AD’s impact. While some dietary components (e.g., B vitamins, ketogenic diet) and physical activity, particularly aerobic exercise, show promise for improving cognitive function and managing symptoms, evidence for consistent benefits remains limited and requires further investigation. Dietary and exercise research in AD faces significant limitations, including intervention complexity, study design challenges, disease heterogeneity, and difficulties in measuring long-term effects. Addressing these limitations is crucial to fully realize the therapeutic potential of these lifestyle interventions in combating AD.

## 1. Introduction

Alzheimer’s Disease (AD), a progressive neurodegenerative disorder, casts a growing shadow on an aging global population in terms of both personal and financial loss [1]. This narrative review delves into the intricate and multifaceted changes in the brain as AD advances, subsequently exploring the associated cognitive impairments and the promising role of modifiable lifestyle factors, specifically diet and exercise, in mitigating its impact. We first dissect the structural, functional, neurochemical, metabolic, and vascular alterations within the brain that are hallmarks of AD, laying the groundwork for understanding the disease’s profound effects on cognition. Following this, we examine the characteristic cognitive decline observed in AD, focusing on early deficits in memory and executive functions and the progression to language and visuospatial impairments. Finally, we turn our attention to the burgeoning evidence highlighting the potential of dietary interventions and physical exercise as powerful tools in managing AD, exploring their effects on brain health, cognitive function, and underlying AD pathology. By synthesizing current research, this narrative review aims to provide a comprehensive overview of the interplay between brain changes, cognitive decline, and the modifiable lifestyle factors, specifically diet and exercise, that hold promise for addressing this devastating disease.

## 2. Changes in the Brain and Cognition Associated with AD

### 2.1. Brain Changes Associated with AD

**Brain structure.** AD involves brain changes like regional atrophy and enlarged ventricles [2,3]. Key hallmarks of AD include amyloid-beta plaques and neurofibrillary tangles (tau protein), which precede neuronal death [4]. Plaques appear first in the neocortex and spread to the allocortex, hippocampus, basal ganglia, midbrain, and cerebellum, years before clinical symptoms appear [5]. Tangles emerge later, starting in the transentorhinal region of the perirhinal cortex and progressing through the hippocampus to the neocortex [6]. The early detection of AD is challenging due to overlapping amyloid and tau pathologies. Advanced brain imaging, including magnetic resonance imaging (MRI), diffusion tensor imaging (DTI), and positron emission tomography (PET) scans using amyloid-beta-specific ligands, aids in identifying microstructural abnormalities and pathological substances, improving diagnostic capabilities [7,8]. In addition, there is also a correlation between AD neuropathological changes and cognitive impairment, thus suggesting that cognitive impairment may correlate with the burden of neocortical neurofibrillary tangles present in the brain [9].

**Synaptic and neurochemical changes.** AD is a progressive, synaptic failure disease whereby synaptic pathology and mitochondrial oxidative damage are early events in AD progression [10]. Loss of synapses and synaptic damage correlate with cognitive deficits found in AD patients. As the disease progresses, there are significant changes at the synapse [11], which result in reduced spine density, impaired memory and coordination of cognitive activities, and reduced signal transmission. Synapse loss occurs early in AD due to soluble amyloid-beta, phosphorylated tau accumulation, and the increased production of mitochondrial-generated free radicals at neuronal synapses [12]. The clinical manifestations of AD result from the impairment of these cerebral pathways, including the basal forebrain, where the cholinergic innervation of cortical areas is the most vulnerable. Specifically, the cholinergic receptors are dysregulated, and impairment of the cholinergic system is considered an early event in AD progression, thereby compromising cognition. The dysregulation of receptors, including other endogenous neurotransmitters, has also been noted in AD in patients and experimental and animal models [13].

Research shows a loss of glutamatergic neurons in AD patients, particularly in the hippocampus’s neocortex and the CA1 region. Additionally, accumulating amyloid-beta and tau proteins result in the death of astrocytes and microglia [13]. Although the dopaminergic system is not traditionally associated with AD [14], a recent meta-analysis linking the dopaminergic system to AD suggests that the level of dopamine and D1 and D2 receptors is decreased in patients with AD. However, the specific role of this neurotransmitter system in AD remains unclear [15]. Overall, early synaptic failure, mitochondrial oxidative damage, and significant changes in multiple neurotransmitter systems, most notably the cholinergic and glutamatergic pathways, leading to cognitive decline, are characteristics associated with AD.

**Brain function.** Normal cognitive function relies on the efficient processing and transmission of information within and between specialized structural regions and functional networks within the brain. AD is portrayed by the disintegration of neuronal connectivity within the brain due to structural changes (i.e., brain atrophy and neuronal dysfunction). Early alterations in brain functional connectivity may be associated with AD pathology [16]. Resting-state functional MRI has shown that as tau spreads through functional connections within the brain, in both amyloid-beta-negative and amyloid-beta-positive individuals, reduced functional connectivity to tau epicenters is observed [17]. Amyloid-beta-PET in tau epicenters has been shown to mediate the association between tau spreading and functional connectivity to epicenters, thus suggesting a partial mediating effect of amyloid-beta deposition in tau epicenters on the local impact of tau spreading on functional connectivity. These findings suggest that tau spreading through connections is locally associated with disrupted functional connectivity between the tau epicenter and non-epicenter regions, independent of amyloid-beta pathology. Amyloid-beta, other co-pathologies, and the apolipoprotein E epsilon 4 (APOE4) allele can all lead to tau-related functional disconnection vulnerability [18]. Overall, AD disrupts normal cognitive function by breaking down neuronal connectivity, primarily through tau protein spreading that weakens functional connections, often independently of amyloid-beta. However, amyloid-beta and other factors can exacerbate this disconnection.

**Metabolic changes**. Metabolic dysfunction during AD is associated with brain glucose hypometabolism that may be observed before the development of many AD symptoms [19]. Insulin resistance, including type 2 diabetes, hyperlipidemia, obesity, or other metabolic disease is also associated with an increased risk for the development of AD [20]. The association between glucose hypometabolism may partially be due to systemic mitochondrial dysfunction [21]. Mitochondria are required for the cellular organelles responsible for the energy production need for neuronal function, and may become impaired in AD, which can result in several cellular consequences [22]. Mitochondrial dysfunction is associated with oxidative stress, the dysregulation of energy metabolism, failures in mitochondrial quality control, and disturbances in calcium release. These abnormalities are associated with neurodegeneration and the development and progression of AD. Mitochondrial abnormalities are early detectable changes in AD pathology [23]. The mitochondrial cascade posits an interconnected relationship between mitochondrial dysfunction and amyloid-beta pathology, creating a vicious cycle that accelerates neurodegeneration [24]. However, this theory’s limitations regarding sporadic AD underscore the need to integrate mitochondrial dysfunction into a comprehensive model of the disease [25]. Mitochondrial-focused approaches represent transformative strategies to combat AD, including metabolic modulators, mitophagy enhancers, antioxidants, and advanced techniques like mitochondrial transplantation and gene therapy [26,27,28,29,30,31,32,33,34,35].

Similar to insulin resistance and mitochondria, there is an association between inflammation and AD [36]. Brain trauma, diet, infections (systemic and local), and the gut microbiota are extrinsic factors impacting AD inflammation [37]. Microglial phagocytosis, blood–brain barrier function, cellular metabolism and cell senescence are intrinsic factors that also play a central role in AD neuroinflammation [38]. Cells such as astrocytes, lymphocytes, oligodendrocytes, peripheral myeloid cells and even vascular cells are activated in AD and contribute to the chronic neuroinflammation that causes a leaky blood–brain barrier [39,40]. Dysregulation of the glymphatic system, which clears brain metabolic waste via cerebrospinal fluid movement, may also contribute to sustained neuroinflammation [41]. In line with the focus on mitochondria, innovative therapeutic strategies designed to target inflammation during AD are presently being assessed in clinical trials.

**Vascular changes.** As with many other brain changes associated with AD, vascular alterations could potentially emerge before the clinical presentation of the disease’s characteristic pathophysiological and cognitive symptoms [42]. Substantial evidence indicates a strong interplay between vascular changes and amyloid pathology in AD [43]. Studies show reduced cerebral blood flow correlating with amyloid accumulation in early AD and gray matter loss in later stages. Vascular health metrics, including cerebral blood flow changes, indicate disease progression, particularly in preclinical populations [44,45]. Age-related vessel deterioration, microvascular abnormalities, and impaired cerebral blood flow, potentially due to amyloid angiopathy and stagnant capillaries, contribute to AD [46,47]. Structural vascular changes, endothelial dysfunction, disruption of the blood–brain barrier, and neuroinflammation play pivotal roles in neurodegenerative pathways [47,48,49,50,51]. Zlokovic’s “two-hit” hypothesis, wherein the first “hit” is initial vascular dysfunction, is proposed to cause a cascade of events that include reducing cerebral blood flow and a compromised blood–brain barrier, thereby hindering the brain’s ability to clear amyloid. This results in “hit 2”, which includes an increase in amyloid accumulation, thus further exacerbating vascular dysfunction [52]. These hits affect neurovascular smooth muscle cells, pericytes, astrocytes, and endothelial cells, and contribute to AD progression [47,51,52]. Impaired angioneurins expression further impacts cerebral blood flow and the integrity of the blood–brain barrier, contributing to neurodegeneration. Improved vascular health may reduce AD progression [52,53,54].

### 2.2. Summary of Pathophysiological Changes Associated with AD

AD involves multifaceted brain changes. Structural changes include regional atrophy, enlarged ventricles, and hallmark plaques/tangles that precede neuronal death. Imaging detects these abnormalities [2]. Functional changes associated with AD disruption in neuronal connectivity, particularly through tau protein spread, lead to functional disconnection. Amyloid-beta and other factors exacerbate this [16,17]. Neurochemical and synaptic changes result in synaptic failure, mitochondrial damage, and neurotransmitter dysfunctions (cholinergic and glutamatergic), which are early events that correlate with cognitive decline [10]. Metabolic changes, including glucose hypometabolism, insulin resistance, and mitochondrial dysfunction, contribute to AD [21]. These factors interact with amyloid and tau pathology [13]. Neuroinflammation, influenced by intrinsic and extrinsic factors, is also crucial [39]. Finally, AD is implicated in cerebrovascular dysfunction, blood–brain barrier disruption, and reduced cerebral perfusion. Vascular changes interact strongly with amyloid pathology, with reduced cerebral blood flow correlating with amyloid accumulation. The “two-hit” hypothesis highlights the interplay between vascular dysfunction and amyloid accumulation [47,52]. In conclusion, AD is a complex neurodegenerative disease characterized by a confluence of structural, functional, neurochemical, metabolic, and vascular alterations that collectively result in progressive cognitive decline. Figure 1 presents a visual summary of brain changes in AD.

### 2.3. Changes in Cognition Associated with AD

Early AD neuropsychological studies sought to define the cognitive profile of mild dementia patients, revealing consistent deficits [55]. A primary finding was a significant episodic memory impairment, specifically in the ability to learn and retain new information [56]. This manifests in everyday challenges like remembering conversations or appointments, and in laboratory tasks involving learning and recalling stories, word lists, or paired associates [57]. Studies comparing AD patients to healthy older adults demonstrated this striking memory deficit [58]. Furthermore, when compared to other forms of dementia, such as frontotemporal dementia or Lewy body dementia, AD patients exhibited a more pronounced difficulty in retaining information over time [59,60].

Beyond memory, early-stage AD patients also showed substantial impairments in executive functions, which encompass abilities like coordinating multiple tasks and shifting between mental sets [61]. These deficits were observed in both individuals with mild and moderate dementia, with evidence suggesting that executive function decline often precedes language and spatial impairments [60,62,63]. Language function is also affected in AD, with semantic memory, the system for processing and storing word meanings, being particularly vulnerable. This is reflected in difficulties with category fluency, naming objects, and making similarity judgments [64,65]. While some studies have reported deficits in word priming, others have not [65]. Visuospatial function, involving spatial reasoning and visual perception, is generally preserved in very early stages, as evidenced by performance in simple copying tasks [66,67]. However, visuospatial impairments become increasingly common as the disease progresses to moderate stages [68]. It is important to note that while these cognitive changes are typical, AD can present with variations. Some individuals experience a gradual decline in spatial abilities, known as posterior cortical syndrome. In contrast, others primarily exhibit language deficits, which can be challenging to differentiate from primary progressive aphasia, a form of frontotemporal lobar degeneration. These atypical presentations highlight the heterogeneity of AD and underscore the need for comprehensive neuropsychological assessments [55]. Perception of cognitive decline is heightened during the early stages of AD, but diminishes later [69]. Overall, early neuropsychological studies of AD consistently identified core cognitive deficits, notably in episodic memory and executive function, with language and visuospatial impairments emerging as the disease progresses. Even so, atypical presentations highlight the disease’s variability.

### 2.4. Behavior, Mood, and Psychiatric Disturbances Associated with AD

AD is associated with a spectrum of neuropsychiatric disturbances, including depression, anxiety, apathy, agitation/aggression, psychosis, and cognitive decline [70,71,72,73]. These symptoms manifest as progressive mood changes, such as increased irritability, sadness, and anhedonia. Behavioral alterations, such as agitation, wandering, and repetitive questioning, are also common. Agitation and aggression often stem from confusion, frustration, or misinterpretation of the environment. Depression and apathy can further accelerate cognitive decline and diminish quality of life. Notably, sleep disturbances, a common feature of AD, contribute to both cognitive and behavioral deterioration. The shared neurobiological basis of these symptoms across neurodegenerative diseases, including AD and Parkinson’s Disease, suggests that similar mechanisms, particularly neurotransmitter dysregulation, underlie these mood and behavioral changes [72].

### 2.5. Changes in Sleep and Sleep Deprivation Associated with AD

In neurodegenerative diseases, including AD, sleep disorders are highly prevalent and negatively impact quality of life for patients and their families. Importantly, a growing body of evidence supports a bidirectional relationship in AD, where disordered sleep acts not only as a clinical symptom, but also as a potential risk factor for influencing the disease’s underlying mechanisms [74,75]. Sleep is essential for clearing brain metabolites, conserving energy, and consolidating memory in the brain. Sleep deprivation has negative effects, such as poor concentration, emotional instability and increased pain sensitivity, and influences metabolic and cardiovascular disease [74,75]. Factors contributing to sleep deprivation include environmental changes, mental health issues, and lifestyle choices. Disruption of the circadian rhythm, such as through shift work, can also negatively influence cognitive performance and overall health.

Sleep and circadian rhythm disruptions are common in patients with AD and can appear during the early stages of the disease [76]. Sleep–wake cycles and circadian rhythms are essential for controlling amyloid-beta levels, as sleep disorders can potentially increase them in the brain [77]. Like amyloid-beta, tau protein levels are also influenced by sleep–wake cycles and significantly increased by sleep deprivation. Thus, tau plays an influential role in neurodegenerative lesions and declines in cognition in AD, with tau pathology possibly preceding amyloid-beta accumulation [78]. Furthermore, the relationship between tau pathology, amyloid-beta, and sleep disorders highlights the need for further research into their interconnected roles in AD progression [79].

### 2.6. Changes in Appetite Associated with AD

Patients with AD sometimes suffer a loss of appetite, which can result in a decrease in body weight [80,81]. While swallowing difficulties and altered appetite are recognized in various neurodegenerative conditions, including vascular dementia and Lewy body dementia, comprehensive studies on eating habits and food preferences in AD remain limited [82,83]. For instance, some individuals with vascular dementia may develop pseudobulbar palsy, leading to dysphagia and a heightened risk of aspiration pneumonia. Similarly, appetite changes are noted in Lewy body dementia, and distinct shifts in food preferences, such as increased appetite and cravings for sweet or pungent foods, occur in frontotemporal and semantic dementias [84,85,86]. However, most research has concentrated on specific eating disturbances like swallowing and appetite. Consequently, there is a notable gap in our understanding of overall eating behaviors and food choices, specifically in **patients with AD** [87].

### 2.7. Summary of Changes in Brain and Cognition Associated with AD

AD is a complex neurodegenerative disorder characterized by a confluence of structural, functional, neurochemical, metabolic, and vascular alterations that collectively contribute to progressive cognitive decline [2]. Structurally, AD involves brain atrophy, enlarged ventricles, and the accumulation of amyloid plaques and tau tangles [8]. Functionally, these alterations disrupt neuronal connectivity, primarily through tau protein spread [2,16,17,18]. Neurochemically, synaptic failure and neurotransmitter dysfunctions are prominent, particularly in cholinergic and glutamatergic systems. Metabolically, glucose hypometabolism, insulin resistance, and mitochondrial dysfunction play significant roles, while neuroinflammation, influenced by intrinsic and extrinsic factors, exacerbates the disease further [24]. Vascular changes, including reductions in cerebral blood flow and disruption in the blood–brain barrier, interact strongly with amyloid pathology, contributing to the progression of AD [40,42,48,52,53]. These pathological changes manifest in cognitive deficits, notably in episodic memory and executive functions, and are accompanied by behavioral, mood, psychiatric, sleep, and appetite disturbances. Early neuropsychological investigations consistently identify core cognitive deficits, while neuropsychiatric symptoms, including depression, agitation, and sleep disorders, further diminish the quality of life [2,73]. Metabolic dysfunctions such as mitochondrial dysfunction and neuroinflammation are also key components of AD [24]. Changes in appetite and weight are also observed. Overall, AD involves a complex interplay of multifaceted brain changes that lead to progressive cognitive decline [2].

## 3. Modifiable Risk Factors in AD

The 2024 Lancet Commission report highlights 14 modifiable risk factors for preventing and delaying AD, 7 of which are associated with lifestyle [88]. Elevated low-density lipoprotein (LDL) cholesterol, hypertension, obesity, type 2 diabetes, depression, sedentary lifestyle, and excessive alcohol consumption are all strongly associated with increased AD risk and can be mitigated through diet and exercise. These factors contribute to the neuropathological changes found in AD, including vascular disease, amyloid accumulation, neuroinflammation, energy dysregulation, and neurotransmitter dysfunction. For example, elevated LDL cholesterol has been linked to increased amyloid plaque formation [89], while hypertension compromises cerebral blood flow and contributes to white matter damage [90]. Obesity in midlife is associated with chronic inflammation and insulin resistance, which can accelerate neuronal dysfunction [91].

Type 2 diabetes increases AD risk through mechanisms such as impaired glucose metabolism and increased oxidative stress [92]. Depression is both a potential early sign of and a factor contributing to AD, potentially due to genetics, AD-related biomarkers elevated in depression, and elevated inflammation that causes vascular damage and weakens the blood–brain barrier [93]. A sedentary lifestyle compounds many of these effects by negatively influencing glycemic control [94] and increasing risk factors such as cardiovascular disease, cognitive decline, and depression [95]. Finally, excessive alcohol consumption is connected to systemic inflammation, reduced brain volumes, and disruptions to neurotransmission [96].

Together, these modifiable risk factors underscore the critical role that diet and regular exercise, which are the focus of this narrative review, play in supporting overall brain health [97,98]. Interventions centered on diet and physical activity not only reduce the prevalence of these conditions, but also offer a robust, multi-modal strategy for lowering AD risk, improving AD symptoms, and slowing cognitive decline. Additional information related to these two adjunction and complementary therapies is detailed within.

## 4. Effects of Diet on Cognition and Brain Health in Patients with AD

Whereas there are a plethora of diets, foods, and natural products that have been investigated for their impacts on brain health in cognitively intact individuals and the prevention of AD [98], a limited number of these dietary interventions have been implemented in trials for individuals with AD. Recent reviews and meta-analyses have focused on a small subset of individual and multi-ingredient vitamins, minerals, fatty acids, and other natural products explored within the context of placebo-controlled, randomized controlled trials (RCT) that have had positive, albeit mixed, effects [99,100]. Conversely, only a few holistic dietary approaches have changed the dietary pattern either in part or entirely in trials for individuals with AD [101,102,103,104].

**Vitamins and Minerals**. Appropriate vitamin and mineral intake is vital for the optimal functioning of the body and the brain. A deficiency in B complex vitamins, for example, can cause elevated levels of homocysteine. Homocysteine has been established as a strong, independent risk factor for Alzheimer’s and related dementias [105]. This is due in large part to its role in cardiovascular disease and AD pathology [106]. Accordingly, there has been extensive research examining the effect of B vitamin supplementation in patients with AD. A 2022 systematic review and meta-analysis examined the impact of B vitamin supplementation on the rate of cognitive decline in 6155 participants across 14 RCTs. The analysis revealed that not only was supplementation associated with a benefit to cognition versus placebo, but in studies where the placebo group showed cognitive decline, vitamin B supplementation slowed cognitive decline for the intervention group [107]. Vitamin D deficiency is also considered a risk factor for dementia. Along with correcting the deficiency, vitamin D supplementation is also suspected to provide neuroprotection via its antioxidant and anti-inflammatory properties. While cohort studies have found a reduced incidence of AD in individuals taking vitamin D supplements [108], intervention studies in patients with AD are mixed, with the majority of studies finding no benefit of supplementation on cognition [109,110,111]. Those studies that have found a benefit often have multiple limitations that make the interpretation of results difficult [112].

Aside from the elevated systemic inflammation associated with age [113], there are many metabolic and cardiovascular risk factors associated with AD that contribute to the inflammatory status of an individual. Therefore, an emphasis has been placed on investigating natural products known to have anti-inflammatory or antioxidant properties. There are several vitamins and minerals with anti-inflammatory and/or antioxidant effects, and have been investigated in RCTs for their ability to reduce AD symptoms and/or slow disease progression. The most promising of these natural products include thiamine, vitamin E, vitamin C, and selenium. A 2022 systematic review concluded that although there was insufficient evidence for the use of vitamin E and C in improving cognition in individuals with AD, thiamine, both alone and with folic acid, had a positive impact on cognition [109]. The literature also suggests that supplementation with selenium improves cognition in patients with AD [114].

**Omega 3-Fatty Acids**. There is a vast scope of literature on the benefits of omega-3 fatty acids in the support of healthy brain aging, due to their role in cell membrane structure and anti-inflammatory activity, and in supporting a healthy vascular system [115]. In observational studies, omega-3 fatty acids are associated with improved cognition [116], but this has not been replicated widely in interventions with patients diagnosed with AD. In a 2020 systematic review and meta-analysis, 38 moderate-quality RCTs provided evidence that omega-3 fatty acid supplementation (with intervention duration at an average of 20.5 months) had little to no effect on new neurocognitive outcomes or cognitive impairment [117]. A 2022 RCT with a sample size of 163 AD patients and an intervention duration of 24 months found that while omega-3 supplementation did not reduce cognitive, functional, or depressive symptom outcomes, there was improvement in the intervention group in sub-items of the ADAS-cog associated with language ability and visuospatial skills [118].

**Other Natural Products.** Several other plant-derived compounds have been evaluated for their ability to improve AD symptoms or slow cognitive decline. Still, few have been implemented in human clinical trials, let alone trials involving AD patients. Curcumin, a polyphenol extracted from turmeric, and the Phyto bioactive compounds in ginseng are considered anti-inflammatory and antioxidant, and have been investigated for their neuroprotective benefits in AD populations [119,120]. The benefits, however, have not been verified, as these studies are limited due to methodological issues, and conclusive evidence has yet to be provided.

**Multi-ingredient Intervention.** Researchers have also considered how multiple vitamins, minerals, and natural products might work synergistically to improve symptoms and disease progression in AD. Although not conducted in AD patients, there is evidence that daily multivitamin–mineral supplementation improves general cognition and episodic memory in older adults over 2 years [121]. Souvenaid, a supplemental drink designed to improve brain function and cognition, has been investigated for its effectiveness in patients with AD. Still, a meta-analysis of four studies concluded that there was no evidence to support its ability to slow the progression of AD, and mixed evidence of its impact on cognition [122].

**Dietary Interventions.** Considering a more ecological approach, researchers have explored how whole diets or dietary patterns might impact symptoms and disease progression in AD. The ketogenic diet has been studied extensively for its ability to shift the body into ketosis, where the use of ketones as a primary energy source has been found to improve brain energy metabolism and cognition [103]. Ketones have also positively affected brain insulin resistance, mitochondrial function, and neurotransmission [123]. In two recent reviews examining a total of 18 RCTs, ketogenic diet adherence was associated with improved general cognition, mental state, and episodic memory in patients with MCI and AD [104]. Supplementing medium-chain triglycerides (MCT oil) can also provide ketones. A recent review found that while some studies have reported improvements in brain energy metabolism, more studies are needed to assess their effects on cognition [124].

Time-restricted eating, such as intermittent fasting, has been investigated for its positive impact on several AD risk factors like cerebrovascular disease and inflammation [125]. This is primarily due to its ability to improve insulin sensitivity, decreasing the risk of type 2 diabetes and obesity. Intermittent fasting also helps the body produce ketones, which is important for minimizing the effects of amyloid-beta and improving cognition. It also improves mitochondrial health and reduces inflammation and oxidative stress, both of which are important for supporting cardiovascular health [125]. Although not in AD, a small yet promising 3-year study of 99 patients with MCI showed that intermittent fasting improved cognitive function, insulin sensitivity, and inflammation [126].

While not as widely studied in AD patients, the Healthy Diet Index and Mediterranean diet were examined in the multi-modal lifestyle intervention MIND-ADmini. The study included an intervention group that received diet education and a supplemental drink, and found a reduced likelihood of declining Clinical Dementia Rating–Sum of Boxes score, but not global CDR scores [101].

### 4.1. Mechanistic Effect of Diet on AD

Specific dietary nutrients are thought to play a key role in modulating amyloid-beta and tau production and mitochondrial dysfunction. The modulation of these factors typically occurs through inflammation and oxidative stress management. As previously mentioned, unsaturated fatty acids, including omega-3 and omega-6, play complex roles in AD, with omega-3 fatty acids generally reducing inflammation and potentially protecting the brain. In contrast, the balance between omega-3 and omega-6 is crucial as they compete for incorporation into cell membranes. DHA, an omega-3 fatty acid, is essential for nervous system function, and has shown promise in reducing amyloid-beta accumulation and inflammation in preclinical studies, suggesting a potential benefit in preventing or slowing early-stage AD. Clinical studies, however, have yielded mixed results, indicating that omega-3 supplementation may be more effective in early stages of the disease or individuals without the APOE ε4 allele, and that considering whole food sources and fatty acid ratios might be more relevant than isolated supplements [115,116,117,118]. Specific vitamins exhibit various neuroprotective effects, including antioxidant, anti-inflammatory, and anti-amyloidogenic activities, as seen with vitamin A reducing tau hyperphosphorylation and influencing amyloid processing, vitamin C and E acting as antioxidants, and B vitamins affecting homocysteine metabolism and DNA methylation. While the evidence suggests the potential benefits of vitamin supplementation in AD prevention and management, clinical studies have yielded inconclusive results, and there are concerns regarding the safety and efficacy of certain vitamin treatments, highlighting the need for further research to determine optimal vitamin interventions and address factors like individual genetic differences and potential toxicity. Moderate alcohol consumption may also be protective through improved cardiovascular health via increased cerebral blood flow and anti-inflammatory properties; however, additional research is warranted. Trehalose, a disaccharide found in various organisms, shows potential benefits in AD by inhibiting amyloid-beta aggregation and rescuing the phenotype of transgenic mouse models. Finally, as indicated previously, elevated levels of LDL cholesterol, often resulting from diets high in saturated fats, can contribute to AD both directly and indirectly by increasing cardiovascular disease risk. Studies in mice and humans suggest that reducing cholesterol levels, through drugs like BM15.766 or statins, may decrease amyloid-beta production and AD prevalence, potentially influencing amyloid precursor protein (APP) processing. While some research indicates that high mid-life cholesterol is a risk factor for later-life AD, and that HDL cholesterol may be inversely correlated with AD risk, the role of cholesterol in dementia is complex, with factors like apolipoprotein E and cholesterol metabolism in the brain also playing significant roles [127]. Specific dietary components, including unsaturated fatty acids, vitamins, and trehalose, and regulating cholesterol levels, may influence AD by affecting amyloid-beta and tau production, mitochondrial dysfunction, inflammation, and oxidative stress. However, the effectiveness and mechanisms vary, and further research is needed to optimize their use.

### 4.2. Summary of the Effects of Diet on AD

Research exploring the effect of dietary interventions on AD reveals a disparity between the plethora of investigated options for cognitively healthy individuals and the limited number tested in those with established AD. Reviews and meta-analyses focusing on specific vitamins, minerals, fatty acids, and other natural products in placebo-controlled trials have yielded mixed results [107,109,112,117,118]. Similarly, only a few holistic dietary approaches have been examined. Regarding specific nutrients, B vitamin supplementation appears promising for slowing cognitive decline in AD patients [107], while the role of vitamin D remains inconclusive despite observational links [112]. Thiamine and selenium have some cognitive benefits, whereas vitamins E and C lack sufficient evidence [109,114]. Omega-3 fatty acids, beneficial for general brain health, have demonstrated little overall cognitive improvement in AD, though one study noted some language and visuospatial benefits [117]. Other natural products like curcumin [119] and ginseng [120] require more rigorous investigation. Multi-ingredient interventions show some promise in older adults without AD, but specific formulations for AD, like Souvenaid, lack substantial evidence [122]. Shifting focus to broader dietary patterns, the ketogenic diet has shown potential for improving cognition and memory in individuals with MCI and AD [104]. Time-restricted eating has also demonstrated cognitive benefits in MCI [125]. While the Healthy Diet Index and Mediterranean diet were explored in a multimodal study, significant cognitive benefits in AD were not consistently observed [101]. Overall, while certain dietary components and patterns offer potential, more robust research is needed to establish effective dietary interventions for individuals living with AD. Figure 2 represents a visual summary of the interventions highlighted above.

## 5. Effects of Exercise on Brain Health and Cognition in Patients with AD

Regular physical activity, including any movement involving skeletal muscles and energy expenditure, contributes to overall well-being. Exercise, a planned and structured form of this activity, specifically targets improvements in physical fitness [128]. Importantly, consistent endurance and resistance training has been shown to reduce the incidence of age-related diseases and mortality, improve risk profiles for chronic conditions, and promote the ability to function independently [129,130,131]. Physical activity, including exercise, has been recognized as a means of preventing and managing AD.

### 5.1. Effects of Exercise on AD

**Prevention.** Growing evidence shows that physical activity and exercise are important for preventing AD [132]. Animal studies indicate that exercise fosters brain health by stimulating neurogenesis [133], enhancing neuronal survival [134], boosting synaptic plasticity [135], and promoting vascularization [136,137]. In healthy older adults, exercise correlates with reduced cerebral amyloid deposition and modulates vascular dementia risk factors [138,139,140]. Specifically, it decreases inflammatory markers and elevates neuroprotective proteins such as BDNF, while also improving glucose metabolism [141,142,143]. Endurance exercise, widely studied for its cognitive benefits, exhibits positive associations with cognitive function and a reduction in age-related brain volume decline in observational and some randomized trials [144,145,146]. Resistance training, although less researched, has demonstrated improvements in executive function, memory, and global cognition, and offers unique benefits to muscle and bone health [147,148,149,150,151,152]. Combined aerobic and resistance training appears to be important for insulin resistance [153,154,155,156,157] and physical function [153,154,156], although there is a lack of direct comparison studies on cognition [158]. Emerging research highlights the potential cognitive benefits of alternative exercises such as yoga [159,160], Tai Chi [161,162], and high-intensity interval training (HIIT) [163,164,165,166,167,168], demonstrating improvements in memory, executive function, and brain structure. However, limitations exist, including the absence of a systematic review, the lack of studies testing current public health exercise recommendations, and the unclear role of alternative exercises in conjunction with traditional forms. Further research is needed to fully understand the independent and combined impacts of various exercise modalities on cognitive function and brain health in older adults.

In addition to prevention studies, low levels of physical activity are a risk factor associated with AD later in life [169]. Findings from population-based cohort studies of older adults who exercise suggest that they have an increased likelihood of maintaining their cognitive function as they age [170]. The English Longitudinal Study of Ageing (ELSA) revealed a significant association between physical activity and the risk of AD. Individuals classified as inactive in this study showed a cumulative AD incidence of 4.8% (95% CI [4.4, 5.4]). At the same time, those with low and moderate-to-high activity levels experienced much lower rates of 0.9% (95% CI [0.8, 1.1]) and 0.2% (95% CI [0.1, 0.5]), respectively. Adjusted analyses indicated substantial risk reductions for AD in the low (60%) and moderate-to-high (78%) activity groups compared to the inactive group. Furthermore, survival analyses over 15 years demonstrated significant differences in dementia incidence across these activity levels. The findings from this study underscore the potential of even light physical activity to play a crucial role in AD prevention [170]. Evidence from a recent meta-analysis (n = 128,261) indicates that engaging in physical activity is associated with a significantly lower risk of AD (RR = 0.86, 95% CI [0.80, 0.93]). This protective association was not significantly influenced by baseline age, the follow-up duration, or the included studies’ methodological quality. Furthermore, dose–response analyses revealed a significant linear and spline relationship, suggesting that greater physical activity may confer greater protection against AD. These findings underscore the importance of physical activity as a modifiable lifestyle factor that can reduce the incidence of AD, even in the long term and independent of reverse causation [171].

**Disease management.** A recent meta-analysis of eight cohort and case–control studies examined whether exercise could improve or maintain functional capacity, physical and cognitive performance, neuropsychiatric symptoms, and quality of life in patients with AD [172]. Six of the eight studies assessed physical function, and five reported positive effects of exercise on physical function tasks post-intervention [173,174,175,176,177]. One intervention even noted that all groups showed declines one year after the intervention. However, the declines were greater for the control group (*p* = 0.003) than for the exercise group [173]. This suggests that exercise could, therefore, have a protective effect. Five of the eight studies assessed cognitive function, and all five reported a positive impact of exercise on cognitive function following the intervention [174,176,177,178,179]. Similarly, two of eight studies examined neuropsychiatric symptoms, and both reported positive effects of exercise post-intervention [177,178]. In addition, two of the eight studies examined the quality of life, and one reported a positive impact of exercise, while the other study reported no difference [174,178]. Similarly, a meta-analysis of 16 trials revealed that physical activity (PA) significantly improved global cognition in AD (SMD = 0.41, *p* < 0.01) [180]. Aerobic exercise (SMD = 0.60) was more effective than mixed exercises (SMD = 0.24). Shorter exercise sessions (<45 min, SMD = 0.66) yielded greater cognitive benefits than longer ones (SMD = 0.27). Moderate to severe AD stages showed larger improvements (SMD = 0.75) than mild-to-moderate stages (SMD = 0.20). The time of the exercise session significantly impacted cognition (β = −0.0105, *p* = 0.03). Nine studies within the meta-analysis indicated that PA also significantly improved Activities of Daily Living (ADL) in AD patients (SMD = 0.56, *p* < 0.001). Other factors, such as exercise duration and frequency, did not show significant differences in cognitive outcomes. Together, these studies suggest that physical activity benefits cognitive function, physical outcomes, and overall disease management in individuals with AD. 

### 5.2. Mechanism of Action for the Effect of Exercise on AD Pathology

**Brain structure.** Studies investigating the relationship between physical activity and brain volume in individuals with AD have yielded mixed results, suggesting a complex interplay between factors. Using tensor-based morphometry imaging, Boyle et al. found that physical activity has a protective effect on brain volume in relation to AD in individuals enrolled in the Cardiovascular Health Study [181]. Higher physical activity levels were correlated with increased overall brain and parietal lobe volume and reduced ventricular dilation, factors often compromised in AD. Conversely, a higher Body Mass Index (BMI) was associated with reduced brain volume, particularly in the frontal, temporal, parietal, and occipital lobes, including the orbitofrontal cortex and anterior cingulate gyrus. Overlapping brain regions, including the orbitofrontal cortex, posterior cingulate gyrus, and posterior hippocampus, are affected by both physical activity and BMI. AD and Mild Cognitive Impairment (MCI) are associated with decreased brain volume, particularly in the frontal lobe and ventricular dilation. While physical activity did not show a direct interaction with AD/MCI diagnosis, BMI did, demonstrating that higher BMI and AD/MCI are associated with reduced brain volume, predominantly in the frontal lobe [181]. Cross-sectional research also shows a significant positive correlation between cardiorespiratory fitness and parietal and medial temporal volume in AD patients. In contrast, non-demented patients did not exhibit a significant relationship between brain volume and cardiorespiratory fitness globally. The association between cardiorespiratory fitness and regional brain volumes in the medial temporal and parietal cortices during early-stage AD suggests a possible protective effect of fitness against AD-related brain atrophy [146]. Overall, very few RCTs have been performed to evaluate the impact of exercise on brain structure in patients with AD. A recent meta-analysis [182] found three studies evaluating the influence of a physical activity intervention on regional brain volume. Morris and colleagues [183] found that individuals with probable AD enrolled in a 26-week RCT comparing the effects of 150 min of aerobic exercise per week versus non-aerobic stretching showed that cardiorespiratory fitness was positively correlated with changes in memory performance and bilateral hippocampal volume (Vidoni et al. [44]). On the other hand, it was found that 52 weeks of aerobic exercise also significantly improved cardiorespiratory fitness (11% vs. 1% in the control group); however, there were no differences in change measures of amyloid, brain volume, or cognitive performance compared to the control group. Similarly, Frederiksen et al. [184] did not find any evidence showing an effect of a 16-week aerobic exercise intervention on brain volume changes in patients with AD. Overall, the impact of physical activity on brain volume and structural changes in patients with AD remains inconclusive.

**Brain function.** Few studies have investigated the influence of physical activity on brain function in AD patients. Several studies have investigated the association between physical activity level and brain function in individuals at risk for AD, specifically in those carrying the APOE4 allele. The findings suggest that higher cardiorespiratory fitness or reported physical activity is associated with greater brain activity when compared to those with lower cardiorespiratory fitness or physical activity levels [185,186]. Brain imaging (functional MRI) has also been used to understand and predict cognitive decline, particularly regarding AD risk and physical activity. Early cognitive decline can lead to increased brain activity, as measured by the BOLD signal, during episodic memory tasks, making it challenging to interpret fMRI results accurately [187]. Paradoxically, higher brain activation during these tasks can predict future cognitive decline. Semantic memory, which encompasses general knowledge, is less affected by normal aging, but is more susceptible to early AD. Lower brain activation during semantic tasks may indicate a higher risk of cognitive decline. Semantic memory areas overlap with brain regions affected by AD, making it a potentially more reliable marker than episodic memory. Studies show that higher PA is linked to greater brain activation during a famous name task, particularly in individuals with APOE4 [188]. This suggests that physical activity may provide neuroprotection and potentially delay cognitive decline in those with genetic risk. In addition, Woodard et al. [188] demonstrated that even among individuals with increased genetic AD risk, high levels of physical activity resulted in brain scans that resembled those of individuals with low genetic risk. Therefore, while the mechanisms are complex and require further study, evidence suggests that physical activity may modulate brain function in individuals at risk for AD, potentially offering a protective effect, particularly in those with genetic predispositions, such as the APOE4 allele.

**Amyloid-beta and Tau.** Exercise may help regulate the production and clearance of amyloid-beta and tau, previously discussed hallmarks of AD. A recent systematic review and meta-analysis using animal models suggests that regular aerobic exercise is associated with a decrease in amyloid-beta [189]. Regular exercise exhibits an anti-amyloid effect in experimental AD models by reducing amyloid-beta. This is associated with a decreased amyloidogenic pathway and an increased non-amyloid pathway, which leads to positive changes in amyloid precursor protein processing through various signaling pathways [189,190]. Similarly, exercise has been shown to reduce brain tau phosphorylation in animal models of tauopathy [191,192]. However, models are limited.

Research exploring the impact of exercise on amyloid-beta in humans is still limited. A meta-analysis of eight studies found no overall effect favoring exercise interventions for both negative (SMD95% = 0.286 [−0.131; 0.704]; *p*  =  0.179) and positive AD status (SMD95% = 0.110 [−0.155; 0.375]; *p*  =  0.416) [193]. The meta-analysis further suggests that exercise interventions do not improve amyloid-beta-related pathology in both healthy individuals and individuals with AD (SMD95% = 0.157 [−0.059; 0.373]; *p*  =  0.155), suggesting that the benefits of exercise for AD are likely related to other mechanistic effects rather than a direct effect on amyloid. Similarly, findings regarding the effect of exercise on Tau in humans have also yielded no significant impact, per a meta-analysis that included four studies [184,194].

**Neurotrophic Factors.** Physical activity stimulates the release of brain-derived neurotrophic factor (BDNF), promoting neuronal growth and survival in mouse models [195,196] and some non-demented older adults [197,198]. However, a meta-analysis by Huang and colleagues included eight studies (7 RCTs and 1 non-RCT) that measured BDNF from blood samples. The results reveal no significant effect of exercise on the ability to change BDNF levels in individuals with AD [194]. Thus, the effect of physical activity and exercise on BDNF in humans is inconclusive.

**Inflammation.** Exercise may reduce inflammation in the brain, which is linked to AD progression [199]. The meta-analysis by Huang et al., which included nine studies (seven RCTs, two non-RCTs), evaluated exercise’s effects on inflammatory factors, notably Tumor necrosis factor (TNF-α) in eight studies and Interleukin-6 (IL-6) in six [194]. The meta-analysis for TNF-α demonstrated an insignificant overall effect, but subgroup analysis revealed a significant decrease in TNF-α levels with aerobic exercise (SMD = −1.21; 95% CI [−2.29, −0.14]). In contrast, the meta-analysis for IL-6 showed a significant reduction following exercise interventions (pooled SMD = −0.45; 95% CI [−0.72, −0.18]) with low heterogeneity (*I*^2^ = 17.0%). No significant changes were observed for other inflammatory factors such as Interleukin-10 or C-reactive protein [194]. These findings indicate a potential link between exercise, reduced inflammation, and AD progression.

**Metabolism.** Participation in physical activity and exercise is associated with improved metabolic indicators and insulin sensitivity [200], and mitochondrial function in older adults [201]. A meta-analysis of 12 studies (10 RCTs, 2 non-RCTs) assessed the effects of exercise on metabolic indicators, including insulin, cholesterol, and cortisol [194]. While eight studies on cholesterol markers (HDL, LDL, triglycerides) indicated a significant decrease in LDL with exercise (pooled SMD = −0.26; 95% CI [−0.50, −0.01]; *I*^2^ = 0.0%), no significant changes were observed for HDL, triglycerides, or total cholesterol [202]. Separately, a study by Ho et al. [202] found that exercise resulted in a more dynamic daily cortisol rhythm compared to the control group, using five saliva samples. Current research also shows that exercise encourages communication between mitochondria and other organelles in AD neurons; however, the therapeutic potential of exercise is not conclusive [203,204].

**Vascular Changes.** Pre-clinical and clinical studies suggest that exercise may enhance cerebral blood flow and vascular function, promoting vascular repair in aging and AD-affected brains [205]. Additionally, gender and APOE status may moderate the cerebrovascular response to exercise in healthy older adults [206], as well as amyloid burden [207]. Unfortunately, limited studies exploring the effect of exercise on cerebrovascular function in patients with AD are available. One small study in 39 patients with AD showed that individuals who completed moderate- to high-intensity aerobic activity and strength training for 6 months experienced increases in flow-mediated dilation (3.725%, *p*  <  0.001), passive leg movement (99.056 mL/min, *p*  =  0.004), the area under the curve (37.359AU, *p*  =  0.037) and vascular endothelial growth factor (8.825 pg/mL, *p*  =  0.004). No difference between pre- and post-treatment was found for any variable in the control group. Additionally, blood flow and shear rate increased during exercise (*p*  <  0.05 for both), but not during the control treatment [175]. These findings suggest that exercise participation may have a positive impact on vascular function in individuals with AD.

**Improved Cognitive Function.** In observational and some randomized trials, exercise in healthy older adults has been shown to positively affect cognitive function [144,145,146,147,148,149,150,151,152]. Regular physical activity has been shown to enhance memory, attention, and executive function in individuals with early-stage AD [208]. A recent review by Demurtas et al. [208] showed that, in patients with AD, physical activity and exercise effectively improved global cognition (SMD = 1.10; 95% CI [0.65, 1.64]). Thus, this review suggests that physical activity/exercise positively affects several cognitive aspects of AD, but additional RCTs are still needed to confirm this relationship [208]. In addition, Liang et al. [209], in a systematic review of 21 studies, also reported improvements in the mini-mental state exam (MMSE) and Alzheimer’s Disease Assessment Scale–Cognition (ADAS-cog) with participation in exercise (MMSE: SMD = 0.46, 95% CI [0.29, 0.63], *p* < 0.01; ADAS-cog: SMD = −0.23, 95% CI [−0.4, −0.06], *p* < 0.01). Regular exercise appears to positively impact memory, attention, and executive function in healthy older adults and individuals with AD, as indicated by improved scores on cognitive assessments. However, further research is needed to solidify these findings.

**Behavioral Symptom Management.** Physical activity serves as a positive non-pharmacological method for alleviating neuropsychiatric symptoms in older adults living with AD [208] Liang et al. [209] showed that exercise participation is associated with improved neuropsychological symptoms (Neuropsychiatric Inventory Questionnaire, NPI: SMD = −0.3, 95% CI [−0.52, −0.08], *p* < 0.01). McCartney et al.’s review of 13 studies [210] also suggests exercise is effective in lessening agitation, highlighting that studies with better exercise adherence showed more significant improvements in agitation and challenging behaviors. Moreover, exercise positively impacts circadian rhythms in healthy older adults, preventing disturbances and aiding resynchronization, contributing to restful sleep [79,211,212]. Previous RCTs in older adults have shown that older adults can successfully improve their sleep quality through exercise [213]. Moderate physical activity has also improves sleep in individuals with AD [214]. Physical activity offers a beneficial non-drug approach for reducing neuropsychiatric symptoms like agitation and improving sleep quality by regulating circadian rhythms in elderly individuals, including those with AD.

### 5.3. Influence of Exercise Type on AD

**Aerobic Exercise**, including activities like walking, jogging, swimming, and cycling, has shown promising results in mitigating the effects of AD. Zhang et al. [215] recently meta-analyzed 15 RCTs. Aerobic exercise significantly improved MMSE scores in AD patients [weighted mean difference (WMD) = 1.50 (95% CI [0.55, 2.45]), *p* = 0.002]. Subgroup analyses revealed that interventions with 30 min sessions [WMD = 2.52 (95% CI [0.84, 4.20]), *p* = 0.003], a weekly duration under 150 min [WMD = 2.10 (95% CI [0.84, 3.37), *p* = 0.001], and a frequency of up to three times per week [WMD = 1.68 (95% CI [0.46, 2.89]), *p* = 0.007] were particularly effective in increasing MMSE scores. Notably, greater cognitive decline at baseline predicted larger MMSE score gains. Unfortunately, aerobic exercise in humans has not been shown to reduce amyloid accumulation in cognitively normal older adults at risk for AD [44] or individuals with AD [193]. **Aerobic fitness and cognitive responses to exercise vary in older adults with AD** [216]. However, additional research is needed to determine the impact of aerobic exercise on AD prevention and the reduction in pathology and associated symptoms.

**Strength Training**. A powerful strategy for physical enhancement, strength training (also known as resistance exercise) involves contracting specific muscles against external resistance. This type of exercise is crucial for building muscle mass and strength, increasing bone density, optimizing body composition, and improving functional capacity and balance. As a result, it can attenuate or even reverse sarcopenia and make task performance easier [217,218]. It may improve cognitive function and overall physical health, which is essential for maintaining independence in individuals with AD. Overall, fewer RCTs utilizing resistance training have been completed compared to those utilizing aerobic exercise. Vital et al. [219] found in a sample of 34 older adults with AD that there was no significant difference associated with cognition in patients with AD when comparing resistance training and social group activities. However, this is a single study. Additionally, although reviews [220] suggest that resistance training may prevent or ameliorate AD, this relationship has not been fully characterized; therefore, additional research is warranted.

**Mind–Body Exercises**. While mind–body exercises such as yoga and Tai Chi are recognized for improving balance, coordination, and cognitive function in older adults [221], the direct impact on AD symptoms is yet to be studied. Nevertheless, research in individuals with MCI indicates that yoga can positively influence sleep, stress, BDNF, serotonin, and brain volume. Since in vitro and in vivo studies have demonstrated the direct action of sleep, stress, and serotonin on amyloid-beta, it is a compelling speculation that yoga and meditation could potentially mitigate AD progression [222]. However, given the lack of RCTs to test this hypothesis, additional research is needed. It can be visualized in Figure 3.

## 6. Exercise and AD Summary

Regular physical activity, particularly structured exercise, is increasingly being recognized for its potential in preventing and managing AD. In prevention, exercise in animal models promotes brain health through neurogenesis, neuronal survival, synaptic plasticity, and vascularization [132,220,223]. Studies in healthy older adults link exercise to reduced amyloid deposition, the modulation of vascular risk factors, decreased inflammation, and increased neuroprotective proteins like BDNF, alongside improved glucose metabolism [44,185,189,190,200,207]. While endurance training shows cognitive benefits and reduced brain volume decline, resistance training improves executive function, memory, and global cognition [44,174,177,183,215]. Combined aerobic and resistance training appears optimal for metabolic and physical function, and emerging research suggests benefits from yoga, Tai Chi, and HIIT, though more systematic research is needed. Population-based studies indicate that even low physical activity levels are associated with a lower risk of developing AD [170,224].

In managing existing AD, exercise has shown promise in improving physical and functional capacity, cognitive performance, and neuropsychiatric symptoms, although effects on quality of life are less consistent. Meta-analyses suggest that physical activity significantly improves global cognition and activities of daily living in AD patients, with aerobic exercise and shorter sessions potentially being more effective [180,182,209,215,225]. Mechanistically, while the impact of exercise on brain structure in AD is still under investigation with mixed results, some studies suggest a protective effect on brain volume and a correlation between cardiorespiratory fitness and brain volume in specific regions [180,215]. Exercise may also modulate brain function, particularly in individuals at genetic risk for AD [185]. While animal studies show exercise can regulate amyloid-beta and tau, human studies have not yet demonstrated a significant direct impact on these AD hallmarks [10,44,189]. However, exercise can influence neurotrophic factors (though not consistently BDNF in AD patients), reduce inflammation (especially TNF-α with aerobic exercise and IL-6), improve metabolic indicators like LDL cholesterol, enhance vascular function, and ultimately lead to improved cognitive function and behavioral symptom management in individuals with AD [175,198,205]. Different types of exercise, such as aerobic, strength training, and mind–body practices, show varying degrees of benefit, warranting further specific investigation [215,219,220,222].

### 6.1. Exercise Recommendations for Individuals with AD

Robust evidence suggests that regular exercise offers protection against AD risk, though the ideal amount is still unknown [225]. Meta-analyses and systematic reviews point to aerobic exercise, specifically sessions around 30 min, less than 150 min per week, and up to three weekly sessions, as beneficial for cognitive function in AD patients. Notably, those with lower initial cognitive function experience more substantial improvements [215,225]. In contrast, the effects of other exercise modalities, including resistance and mind–body exercises, are not yet supported by the available evidence.

### 6.2. Diet and Exercise, Limitations of Available Research

Dietary research on AD is hampered by several inherent limitations. The complexity of nutritional interventions, involving numerous interacting components, contrasts with the simplicity of single-compound drug trials, making it challenging to pinpoint specific beneficial elements. The typically slow progression of AD necessitates lengthy and costly studies to observe meaningful effects, often plagued by difficulties in maintaining participant adherence and high dropout rates. The heterogeneity of AD, with its diverse presentations influenced by individual factors, further complicates the identification of universally effective dietary strategies. Blinding participants and researchers to dietary changes is often impractical, introducing potential bias. Accurately assessing long-term dietary intake remains a methodological challenge, relying on potentially inaccurate recall-based methods. Many current studies suffer from small sample sizes, limiting their statistical power. The timing of intervention relative to the disease stage is critical but often variable across studies. Participant-related factors, such as advanced age, co-existing health conditions, cognitive impairment affecting eating habits, and varying socioeconomic circumstances influencing access to specific foods and support, add further layers of complexity. Finally, the sensitivity of cognitive tests to detect subtle dietary-induced changes and the limited immediate responsiveness of AD biomarkers pose challenges in measuring outcomes. Overcoming these limitations through more rigorous study designs, larger and more diverse cohorts, extended intervention durations, improved assessment methods, and relevant outcome measures is essential to advance our understanding of the role of diet in addressing AD.

Similar to the research on diet, the research on the effects of exercise on brain health and cognition in the context of AD also faces several limitations. While animal studies offer promising insights into the mechanisms by which exercise might prevent AD, including promoting brain health at a cellular level and potentially regulating amyloid and tau, these findings have not consistently translated to human studies, particularly concerning direct impacts on AD’s biological hallmarks. The effects of exercise on brain structure in individuals with AD also remain unclear, with studies yielding mixed results. Furthermore, the influence of exercise on neurotrophic factors like BDNF in AD patients is not consistently observed. Research on alternative exercise types beyond aerobic and resistance training, such as yoga, Tai Chi, and HIIT, is still emerging and requires more systematic investigation. A lack of direct comparative studies between different exercise modalities and the need for more robust RCTs further limit the current understanding. Establishing optimal exercise parameters and consistently demonstrating improvements in quality of life for AD patients also present ongoing challenges. The complexity of interpreting brain function changes in early AD and the limited research on cerebrovascular function in this population add to the existing limitations. Finally, the potential role of mind–body exercises in managing AD symptoms warrants dedicated future research [172].

Finally, we would like to address the limitations of the current narrative review. The purpose of a narrative review is to provide a broad overview of a topic, in this case, the effect of diet and exercise on AD. The methodology for narrative reviews is flexible and often not explicitly defined; the literature selected is based on our expertise and may be subjective. In addition, the literature is only narratively described, and data were not synthesized. A quality assessment of the studies cited is also not provided. Thus, bias may be present. Even so, this review provides a thoughtful summary of the current state of the literature describing the effects of dietary and exercise interventions on AD and can be visualized in Figure 4.

## 7. Conclusions

In conclusion, this narrative review has illuminated the intricate cascade of structural, functional, neurochemical, metabolic, and vascular alterations that characterize the AD brain, culminating in a progressive decline in cognitive abilities, particularly episodic memory and executive functions, and often accompanied by behavioral, mood, sleep, and appetite disturbances. Furthermore, it has explored the significant role of modifiable lifestyle factors, specifically diet and exercise, in potentially mitigating the impact of this devastating disease. While research into specific dietary components and holistic dietary patterns offers some promising leads, particularly regarding B vitamins and the ketogenic diet, the evidence for consistent benefits in established AD remains limited and requires further rigorous investigation. Similarly, physical activity, especially aerobic exercise, demonstrates potential in improving cognitive function, managing neuropsychiatric symptoms, and possibly influencing underlying AD pathology. However, the precise mechanisms and optimal exercise parameters are still being elucidated. Despite the encouraging findings, dietary and exercise research in the context of AD faces considerable limitations, including the complexity of interventions, study design and adherence challenges, the heterogeneity of the disease, and difficulties in accurately measuring long-term effects on cognition and brain biology. Recognizing and addressing these limitations in future research endeavors is crucial to fully harness the therapeutic potential of these modifiable lifestyle factors in the ongoing fight against AD.

## Figures and Tables

**Figure 1 nutrients-17-01804-f001:**
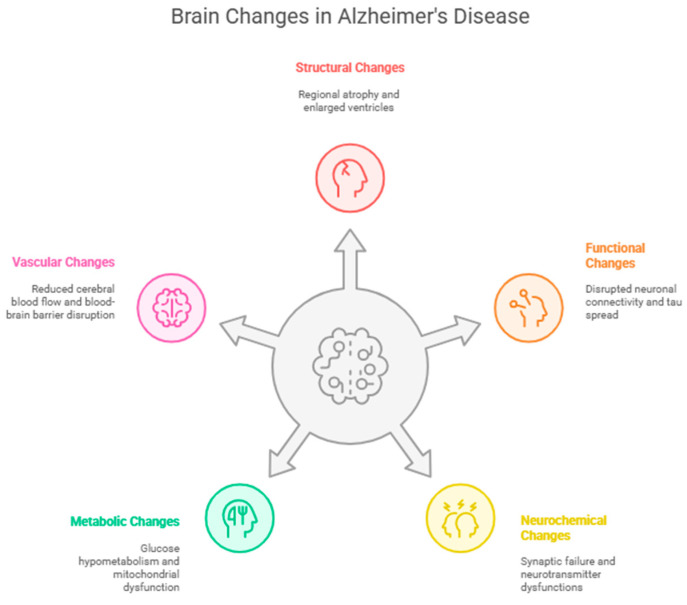
Brain changes in Alzheimer’s Disease.

**Figure 2 nutrients-17-01804-f002:**
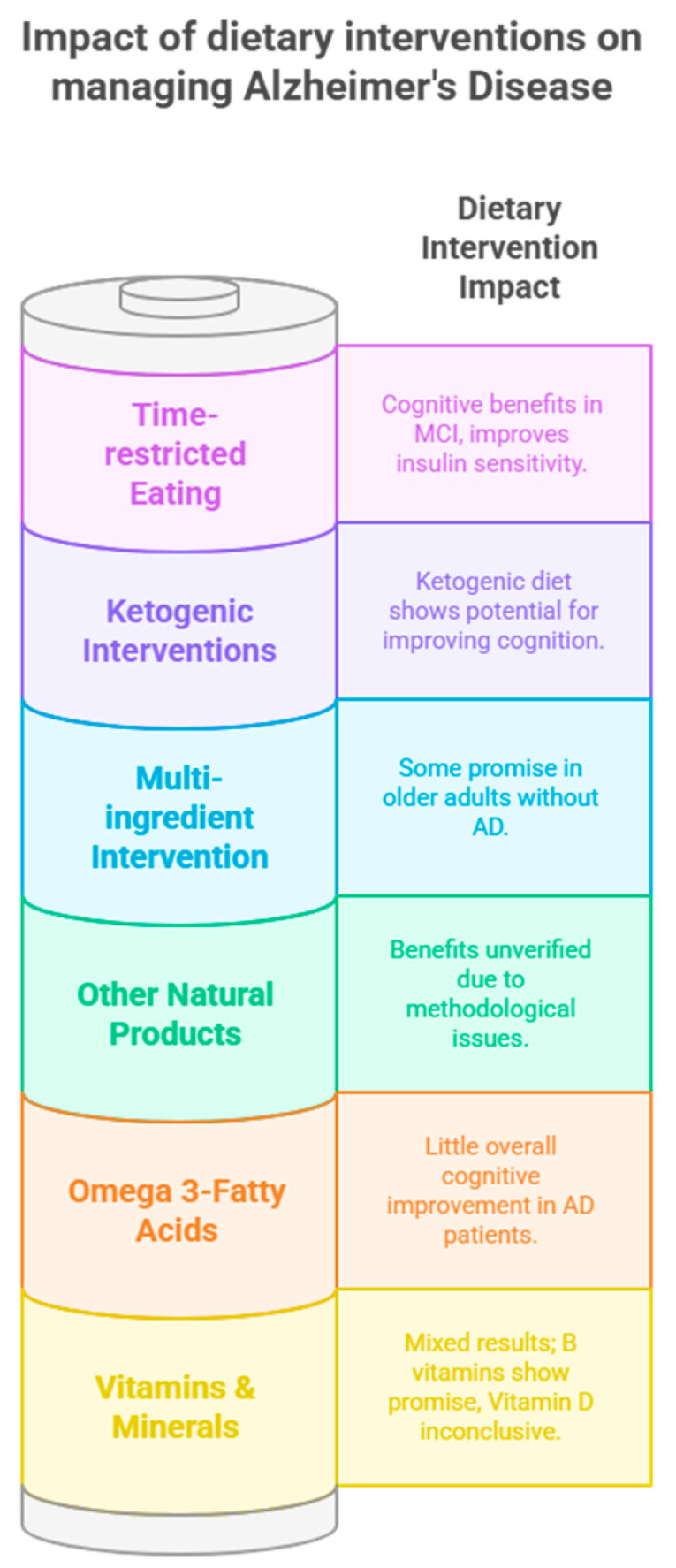
Impact of dietary interventions on managing Alzheimer’s Disease.

**Figure 3 nutrients-17-01804-f003:**
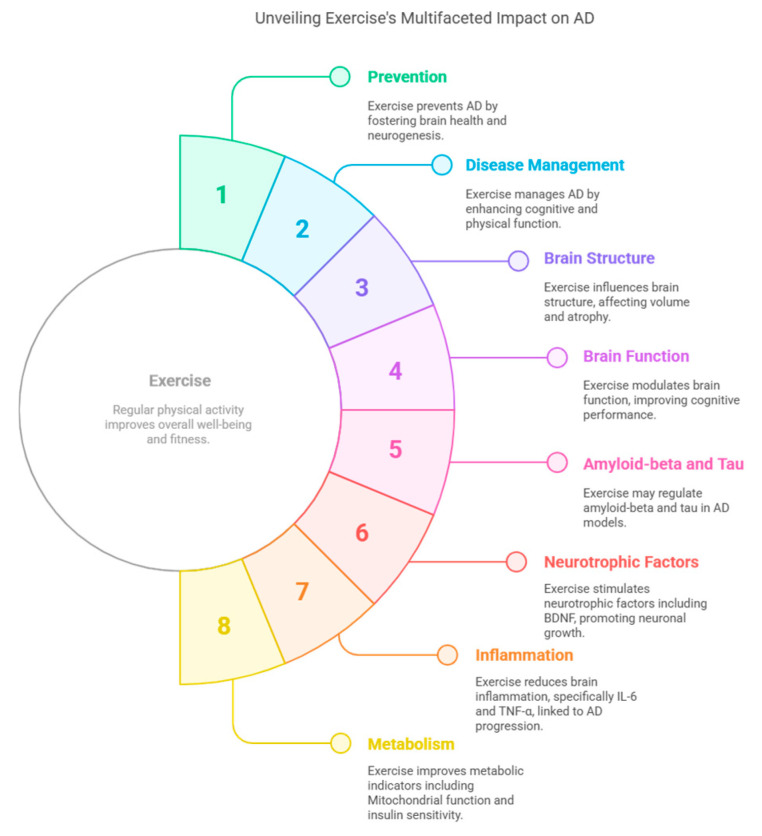
Unveiling exercises multifaceted impact on AD.

**Figure 4 nutrients-17-01804-f004:**
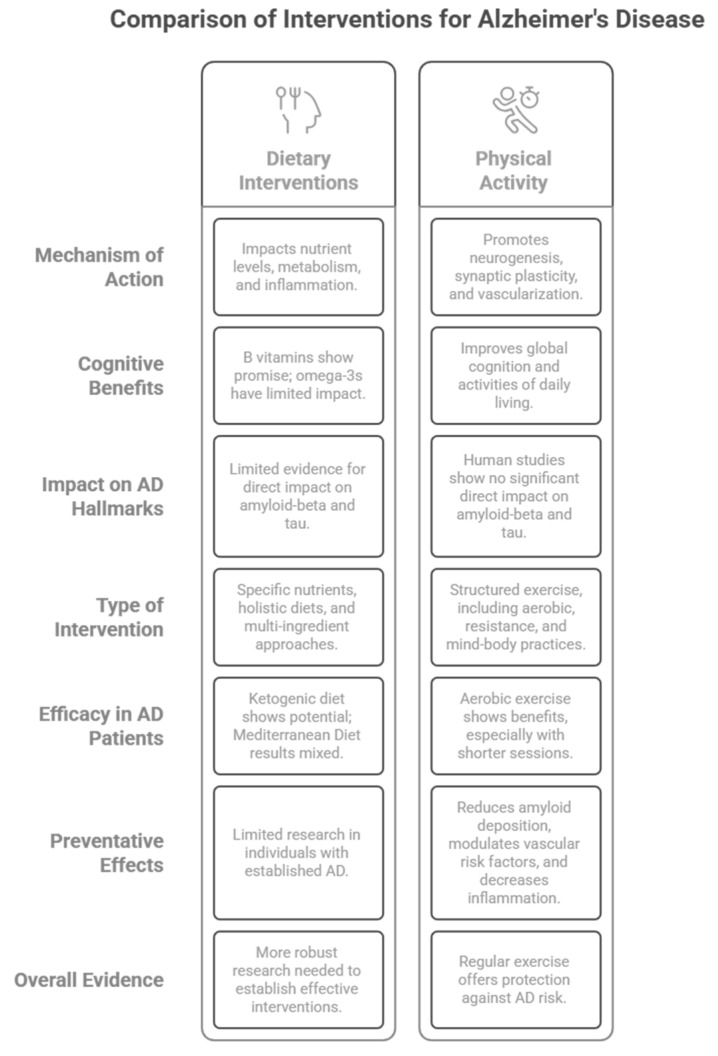
Comparison of interventions for Alzheimer’s Disease.

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
