# Peer review of "A Narrative Review Evaluating Diet and Exercise as Complementary Medicine for the Management of Alzheimer’s Disease"

_nutrients, 2025, doi:10.3390/nu17111804_

Round 1

Reviewer 1 Report

Comments and Suggestions for Authors

The review study conducted by Szabo-Reed and Key evaluated the effect of diet and exercise as modifiable lifestyle factors for the control and prevention of Alzheimer's disease. The study presents important data for the field, in addition to being innovative and well-conducted. However, it is important that the authors present:

- A topic involving cellular mechanisms that participate in the positive results found in diet and exercise against Alzheimer's disease;

- A figure that summarizes all the findings of the review.

Author Response

The review study conducted by Szabo-Reed and Key evaluated the effect of diet and exercise as modifiable lifestyle factors for the control and prevention of Alzheimer's disease. The study presents important data for the field, in addition to being innovative and well-conducted. However, it is important that the authors present:

  • A topic involving cellular mechanisms that participate in the positive results found in diet and exercise against Alzheimer's disease;

Thank you for this suggestion. A paragraph detailing the potential mechanistic effect of diet on AD has been added to page 9. The cellular impact of exercise on alzheimers disease is disucssed in section 5.2., Metabolism. Here the impact of exercise on mitochondrial function is discussed as well as the impact of exercise on other metobolic factors. 

  • A figure that summarizes all the findings of the review.

Four figures have been added to summarize the findings of this review.

Reviewer 2 Report

Comments and Suggestions for Authors

This manuscript addresses an important topic at the intersection of nutrition, exercise, and Alzheimer’s disease. The breadth of literature covered is impressive and the paper is well referenced. However, the manuscript currently reads more like an annotated bibliography than a critical review. Major improvements are needed in terms of methodological rigor, synthesis of results, and clarity of writing. The lack of critical appraisal and inadequate methodological detail limit the utility and credibility of the paper in its current form. It has the potential to make a meaningful contribution if these issues are addressed.

  1. The manuscript lacks a clear and concise research objective or hypothesis. The introduction is extensive but does not pinpoint the specific gap this review aims to address.
  2. The methodology section does not provide sufficient details on the selection criteria, search strategy, or quality assessment of the included studies.
  3. The results section includes a wide variety of interventions (nutritional, physical exercise, lifestyle) without sufficient critical synthesis. There is excessive listing of individual study findings without clear thematic integration.
  4. The manuscript heavily cites existing meta-analyses and systematic reviews but contributes little novel synthesis or critical evaluation.
  5. The conclusion is too general, simply stating that lifestyle factors affect Alzheimer’s disease without specifying which interventions show the strongest evidence.

Author Response

This manuscript addresses an important topic at the intersection of nutrition, exercise, and Alzheimer’s disease. The breadth of literature covered is impressive and the paper is well referenced. However, the manuscript currently reads more like an annotated bibliography than a critical review. Major improvements are needed in terms of methodological rigor, synthesis of results, and clarity of writing. The lack of critical appraisal and inadequate methodological detail limit the utility and credibility of the paper in its current form. It has the potential to make a meaningful contribution if these issues are addressed.

  1. The manuscript lacks a clear and concise research objective or hypothesis. The introduction is extensive but does not pinpoint the specific gap this review aims to address.

As stated on page 1, line 36, "By synthesizing current research, this narrative review aims to provide a comprehensive overview of the interplay between brain changes, cognitive decline, and the modifiable lifestyle factors, specifically diet and exercise, that hold promise for addressing this devastating disease."

2. The methodology section does not provide sufficient details on the included studies' selection criteria, search strategy, or quality assessment.

The review presented is a narrative review, not a systematic review. Although narrative and systematic reviews aim to synthesize the existing literature, they differ significantly in their methodology, scope, and rigor. We selected a narrative review to provide the reader with a broad overview of alzheimers disease and how the course of the disease may be impacted by lifestyle, particularly diet and exercise. The methodology for a narrative review is flexible and not explicitly defined, unlike the methodology required for a systematic review. In addition, a narrative review does not require a quality assessment of the literature.  

3. The results section includes a wide variety of interventions (nutritional, physical exercise, lifestyle) without sufficient critical synthesis. There is excessive listing of individual study findings without clear thematic integration.

Thank you for this critique. We apologize that you did not find this narrative review cohesive. In an attempt to improve thematic integration, we added three figures. Figure 1 summarizes brain changes that occur due to AD, Figure 2 summarizes the impact of dietary interventions on managing AD, and Figure 3 describes the impact of exercise on AD. Figure 4 has been added to summarize the two intervention types together.

4. The manuscript heavily cites existing meta-analyses and systematic reviews but contributes little novel synthesis or critical evaluation.

The purpose of this manuscript is to provide a narrative review and describe the available research. Therefore, the purpose is to evaluate the quality of the literature available. We did, however, provide some notable limitations within the literature that should be addressed by future studies. 

5. The conclusion is too general, simply stating that lifestyle factors affect Alzheimer’s disease without specifying which interventions show the strongest evidence.

Thank you for this critique. We would love to state which factors have the strongest level of evidence; unfortunately, as noted in the limitations, the heterogeneity of the literature available makes it challenging to make such claims. This and some other limitations associated with narrative reviews have been added to the limitations section on page 19, line 805. In addition, figure 4 has been added to describe the current interventions and their effects on AD.